# Dataset of Submerged Sand Deposits organised in an interoperable Spatial Data Infrastructure (Western Sardinia, Mediterranean Sea)

Walter Brambilla[1], Alessandro Conforti[1], Simone Simeone[1], Paola Carrara[2], Simone Lanucara[2] and Giovanni De Falco[1]

[1]Istituto per lo studio degli impatti Antropici e Sostenibilità in ambiente marino del CNR, Oristano, Italy

[2] Istituto per il Rilevamento Elettromagnetico dell'Ambiente CNR, Milano, Italy

*Correspondence to*: Walter Brambilla (walter.brambilla@iamc.cnr.it)

**Abstract.** The expected global sea level rise by the year 2100 will determine an adaptation of the whole coastal system and the land retreat of the shoreline. Future scenarios coupled with the improvement of mining technologies will favour increased exploitation of sand deposits for nourishments, especially for urban beaches and sandy coasts with lowlands behind. Objective of the work is to provide useful tools to support planning actions in the management of sand deposits located on the continental shelf of western Sardinia (western Mediterranean Sea). The work has been realised through the integration of data and information collected during several projects. Available data consist of morpho-bathymetric data (multibeam) associated with morphoacoustic (backscatter) data, collected in the depth range -25 to-700 m. Extensive coverage of high-resolution seismic profiles (Chirp 3.5 kHz) have been acquired along the continental shelf. Also, surface sediment samples(Van Veen grab and box corer) and vibrocores have been collected. These data allow mapping of the submerged sand deposits with the determination of their thickness and volumes, and their sedimentological characteristics. Furthermore, it is possible to map the seabed geomorphological features of the continental shelf of western Sardinia. All the available data (doi: https://doi.org/10.1594/PANGAEA.895430) have been integrated and organised in a geo-database implemented through a GIS and the software suite Geoinformation Enabling Toolkit StarterKit ® (GET-IT), developed by researchers of the Italian National Research Council for RITMARE project. GET-IT facilitates the creation of distributed nodes of an interoperable Spatial Data Infrastructure (SDI) and enables unskilled researchers from various scientific domains to create their own Open Geospatial Consortium (OGC) standard services for distributing geospatial data, observations and metadata of sensors and datasets.

Data distribution through standard services follows the guidelines of the European Directive INSPIRE (DIRECTIVE 2007/2/EC); in particular, standard metadata describe each map level, containing identifiers such as data type, origin, property, quality, processing processes to foster data searching and quality assessment.

**Copyright statement**

## 1 Introduction

The sediments used for coastal defence from erosion and flooding have assumed a strategic value in the last decades and they will be even more important in the future considering the expected scenarios of sea level rise.

Several studies have predicted that the global sea level will rise at 2100 in a range between 530 and 1400 mm (Church 2013; Horton et al., 2014; Kopp et al., 2016; Mengel et al., 2016; Rahmstorf et al., 2007). The predicted sea level rise can determine an adaptation of the whole coastal systems in terms of shoreline retreat and growing risk of coastal flooding (Antonioli et al., 2017). In order to counteract the coastal flooding due to sea level rise, an increasing in the exploitation of Submerged Sand Deposit (SSD) could be necessary, as a strategic resource to carry out coastal protection programmes.

The management of coastal erosion has received greater attention during the past decade (Jimenez et al., 2011). Several projects funded by the European Commission (Eurosion, CONSCIENCE, Micore) were launched to achieve the strategic objective of developing guidelines on sustainable management of coastal erosion in Europe (Marchand et al., 2011). A basic concept of coastal erosion management is the 'favourable sediment status', defined by Eurosion project as the situation where the availability of coastal sediments supports the objective of promoting coastal resilience in general and of preserving dynamic coastlines in particular (Merchand et al., 2011, Gault et al, 2011, Sànchez-Arcilla et al., 2011).

The Eurosion concepts were applied in the CONSCIENCE Project that includes on the sediment cell also the sediment reservoirs that can act as a source of sediment (Van Rijn, 2010). These aspects are also discussed on MICORE project that stressed on the application of nourishment projects to contrast climate change effects, in particular, the extreme storms and the sea level rise (Ciavola et al., 2011).

Within this framework, the strategic sediment reservoirs are an essential component and they can be used as sediment supply for nourishment as an intervention measure.

The exploitation of SSD can be addressed to: (i) beach nourishment, especially along urban beaches and sandy coasts facing lowlands, and (ii) the reconstruction and recovery of the foredune.

Extensive interventions of beach nourishment based on the massive use of sediments have been carried out in northern and southern Europe and along the United States to tackle problems of erosion (Finkl and Walker 2005; Radermacher et al., 2017; Pranzini 2017; van Egmond et al., 2018). In some cases, these interventions were realised using tens of millions of $m^3$ of sand (Richards et al., 2009). Due to the large volumes of sand required for beach nourishment projects, dredging from the offshore sediment reservoirs is the preferred method of sediment supply. As easily accessible and previously known deposits have been exploited, apparent sand reserves are seen as a dwindling resource that becomes more precious over time (Finkl and Khalil, 2005). In fact, the marine sand deposits on the continental shelves are a non-renewable resource. For these reasons, some countries have implemented mapping and characterization of SSD in order to plan their future exploitation: the characterization of SSD, in terms areal distribution, thickness, grain size, chemical and mineralogical composition, represents the first step of knowledge to plan the exploitation of reservoirs over the next decades, considering the predicted sea level rise scenarios.

Along the Mediterranean Sea, and in particular along the Italian coastline, the characterization of SSD is one of the main objective of the Italian Ministry of Environment Land and Sea that in 2017 issued national guidelines to prevent and protect coasts from erosion and from the effects of Climate Changes (Eurosion, 2004; MATTM – Regioni, 2018).

There are several examples all around the world on the utilization of SSD for the nourishment projects (Finkl et al 2005). In Italy, the Emilia-Romagna Region developed a geodatabase named IN_SAND, that help the managers to plan the SSD exploitation (Correggiari et al. 2012 and 2016). It is also known that the dredging operation can negatively affect the marine ecosystems and in particular the sessile benthic communities, (Rogers, 1990; Desprez, 2000; Erftemeijeret al., 2012; Fraser et al., 2017). For this reason, accurate procedures aimed to minimize the impact of nourishment operations (including dredging) are adopted by several countries and must to be required when a nourishment project is developed (Finkl et al., 2005; Radermacher et al., 2017). Along the Mediterranean Sea, this operation can affect the *Posidoniaoceanica* meadows. In Italy, for this reason, ISPRA and CNR develops the ENV_SAND database (Grande et al., 2015) aimed to collect the data acquired during the monitoring programme of a nourishment project, including the dredging of the SSD (Nicoletti et al., 2018; MATTM-Regioni, 2018). The introduction of these geodatabases provide instruments that could be very helpful on the management of the SSD not only in terms of sand volumes but useful to (i) support the procedures of environmental impact assessment, (ii) monitor the marine environmental condition during the activities and (iii) manage the impact of dredging and nourishment activities on marine ecosystems (Erftemeijeret al., 2012; Fraser et al., 2017; Nicoletti et al., 2018; MATTM-Regioni, 2018).

 In this work, we present a large dataset collected and organised to support the management of the SSD located along the continental shelf of western Sardinia island (Mediterranean Sea).

The dataset results from the integration of geomorphological, seismic and sedimentological data collected in several projects, along with the western Sardinian (Italy) margin. A set of maps showing information about thickness, volumes and sedimentological features of SSDs have been produced. They represent a fundamental knowledge to support planning and management actions of maritime space and resources. The dataset has been organised following the indication of InSand guide line (Correggiari et al., 2016) issued by the Italian Ministry of Environment Land and Sea . Furthermore, all data have been organised and are managed by means of the software suite Geoinformation Enabling Toolkit StarterKit ® (GET-IT), developed by the Italian National Research Council, which facilitates the creation of distributed nodes of an interoperable Spatial Data Infrastructure.

## 2 Regional setting

The sand reservoirs are located along the shelf of the central sector of the western Sardinian margin (western Mediterranean Sea, Fig.1), a passive type margin, tectonically stable since late Quaternary (Lambeck et al., 2011). The morphology of the margin is mainly controlled by tectonic features and is characterised by a wide amphitheatre facing the Gulf of Oristano, which shows a smooth transition between the 25 km wide continental shelf and the deep basin. The Oristano amphitheatre is

bordered to the north and south by a continental shelf which is twice as wide (Sage et al., 2005; De Falco et al. 2015). Along the study area, the Paleozoic–Early Miocene acoustic basement rises and outcrops (Fig.1). Eastwards, a small basin was filled by the Pliocene sequence and westwards the basement is depressed and is overlaid by the Pliocene and Quaternary sequence (Lecca, 2000; De Falco et al. 2015). South of the study area, the acoustic basement is tectonically down-thrown

and the resulting structural depression is filled by the Miocene sequence, truncated at the top by the Messinian erosional surface and covered by the Pliocene sequence (Lecca et al., 1986).

The continental shelf is sediment-starved, with limited land to sea sedimentary run-off. The transgressive deposits associated with the last sea level rise are characterised by siliciclastic sands and calcareous bioclastic sands (Carboni et al., 1989; De Falco et al., 2008; De Falco et al., 2010, De Falco et al., 2015, De Falco et al., 2017) along the inner shelf as well as clays

and calcareous mud along the outer shelf (Carboni et al., 1989).

Along the wave-dominated and starved shelves, transgressive deposits mainly derive from the re-working of sediments deposited during either low stand conditions of the sea level or the production and re-working of intrabasinal biogenic carbonate sediments (Emery, 1968; De Falco et al., 2015).

The western coast of Sardinia is characterised by a low degree of urbanization and extended pristine coastal sectors. The

sandy shores of this sector are extremely varied, including long linear beaches, wide large transgressive dune fields, embayed beaches and barrier-lagoon systems (Simeone and De Falco, 2012; De Falco et al., 2014). Beach sediments are heterogeneous in composition (terrigenous vs. biogenic carbonate) and grain size (De Falco et al., 2014). Starting from the south of the studied area (Fig.1) wide linear multibarred beaches, with large parabolic dunes, characterise the open coastline between Cape Pecora and Cape Frasca, with a few embayed beaches. Foreshore sediments are mainly terrigenous and are

characterised by coarse grain size, ranging from slight gravelly to gravelly sands (Arisci et al., 2003; De Falco et al., 2014; PueyoAnchuela et al., 2017). In the Gulf of Oristano, the morphology is characterised by the presence of a foredune and the absence of bars in the shoreface. The sediment composition is terrigenous, with the exception of the northern sector of the gulf where sediments are mixed (Tigny et al., 2007; Simeone et al., 2012; De Falco et al., 2014) Grain size ranging from fine sand to gravelly sand. The coastal sector located between Cape Mannu and Cape San Marco is characterised by wide mixed

bioclastic and siliciclastic beach (in the proximity of Capo Mannu), coarse siliciclastic beaches (Sinis Peninsula) and, in the proximity of San Marco, embayed and semi-embayed mixed bioclastic terrigenous beaches.  Grain size vary from medium to coarse sand in the proximity of Cape Mannu, from coarse sand to very fine gravel along Sinis Peninsula and from fine sand to very coarse sand in the proximity of Cape San Marco (Simeone et al., 2018; De Falco et al., 2017; De Falco et al 2014; De Falco et al., 2003).

## 3 Methods

The thicknesses and volumes of the submerged sand deposits and their sedimentological characterisation were obtained through integration and analysis of available data collected during several projects (MAGIC, RITMARE, SIGLA). Geophysical data were collected during several oceanographic cruises using the CNR research vessels R/V URANIA, R/V MINERVA UNO, R/V THETIS. The sector where the SSD is located is showed in Fig.1. The organisation of the GIS of SSD was based on the GIS architecture developed by Correggiari et al. (2016) for the SSD of the Emilia Romagna Region (InSAND project).

To ease processing and data sharing among researchers, all available data were integrated and organised in a geodatabase implemented through a GIS and the software suite Geoinformation Enabling Toolkit StarterKit ® (GET-IT), (Fugazza et al., 2014; Pavesi et al.,2016; Lanucara et al., 2017) developed by researchers of the Italian National Research Council for RITMARE project. GET-IT facilitates the creation of distributed nodes of an interoperable SDI and enables unskilled researchers from various scientific domains, such as geologists, oceanographers, and biologists, to create their own Open Geospatial Consortium (OGC) standard services for distributing geospatial data, observations and metadata of sensors and datasets in an interoperable way. Interoperable SDI approach prevents useless and error-prone duplication of data sources, which are made available (for visualization, access and, if allowed, download) through standard web services, reachable from whichever OGC standard web authorised client. The following subsections illustrate methods adopted for different data collected in the work.

### 3.1 Multibeam echosounder data

Multibeam echosounder (MBES) data were collected along 2400 km$^2$ of the western continental shelf of Sardinia (Fig. 2)in order to create a detailed bathymetric map, thus providing a Digital Terrain Model (DTM) of the seabed along the mid-outer continental shelf/upper slope sectors. Different MBES devices were used. They include a Reson SeaBat 7125 operating at a sonar frequency of 200 – 400 kHz, Reson SeaBat 8111 (100 kHz), Kongsberg EM710 (100 kHz) (see De Falco et al., 2010, 2015, 2017 for a detailed description of MBES and backscatter data acquisition and processing). MBES data allowed the production of the seabed DTM at a resolution of 2.5 m (Fig.3A). Furthermore, a backscatter map (Fig.3B) of relative backscatter intensity values was realised to recognize the morphological features of the seabed. Rocky outcrops and superficial limits of sedimentary bodies were mapped through the interpretation of the DTM with 2.5 m cell size and backscatter map. MBES data were processed using the software Reson PDS-2000 and DTMs were produced and analysed by using the Golden Software Surfer and Global Mapper software.

### 3.2 Seismic data

More than 100 km of very high-resolution seismic data were collected along the study area (Fig. 2) using a Data Sonic Chirp II operating at 3.5 kHz (2.5-7 kHz). The base of SSD was determined from reflection characteristics visible from profiles that were interpreted by means of the GeoSuite® software and calibrated with the vibrocore data.

More than 100 km of high-resolution, single-channel seismic profiles were collected in the investigated shelf sector. Thetechnical specifications and the methods of seismic data processing are described in De Falco et al. (2015).

The interpretation of seismic profiles is based on the seismic stratigraphic criteria of Mitchum et al. (1977) and is mostly in accordance with former interpretations reported in the literature on the regional geology (Fais et al. 1996; Lecca 2000; Casula et al. 2001; Conforti et al., 2016). The seismic data analysis and the stratigraphic features of the study site have been

described in De Falco et al., (2015) and Conforti et al., (2016).

The maps of seismic facies boundaries have been used as key-beds to define the lower limit of sand reservoirs, called Sand Base (Fig. 4). The Sand Base is the unconformity which separates the transgressive sands deposited during the Holocene from the underlying Pleistocene sediments or Miocene to Pliocene bedrock.

The shape of the Sand Base surface was obtained by interpolating the depth values (referred to the present sea level) of the

surface, from each seismic lines, in order to obtain the digital terrain model 10 m cell size of the bottom of sand deposits.

The conversion of two-way travel time to real depth was obtained assuming an average velocity of about 1,550 ms$^{-1}$ within the first 300 ms of the seismic record below the sea floor (Carlson et al. 1986; Budillon et al. 2011).The vertical resolution of seismic data is 0.1 m.

### 3.3 Sedimentological data

Sediment samples collected during the oceanographic cruises (Fig. 2) cited above were used to characterise the sandy sedimentary deposits.  149 superficial sediment samples were collected with a Van Veen grab and a Box Corer, whereas 5 sediment cores up to 3 m long were collected by using a vibrocorer (Fig. 2). In all samples, grain-size distribution was measured using dry sieving for the gravel/sand fraction between 4000 and 90 μm at half-phi intervals. Multivariate statistical techniques were used to classify sediment samples into sedimentary facies (De Falco et al., 2015; De Falco et al., 2011;

Brambilla et al., 2016; De Muro et al., 2017). The statistical analysis of sediments data was performed by the software Gradistat (Blott and Pye, 2011). Sedimentological data provided the ground-truth observations used to classify the acoustic facies obtained by multibeam bathymetry and backscatter data.

### 3.4 Thickness and volumes computation

The model of the thickness of SSD was created with the GIS spatial analysis tools. The thickness was estimated by

subtracting the elevation value of each cell of the DTM of the Sand Base surface from the corresponding cell ofthe DTM of the seabed obtained through MBES surveys. The latter DTM was reduced from 2.5 m to 10 m cell size. The volumes were

estimated from the digital map of the thickness of each deposit (Fig.5) (Correggiari et al.,2013; Correggiari et al., 2016; Trobec et al., 2018).

The submerged sand deposits recognised were classified in seven hierarchic categories, based on the volume of each deposit as shown in table 1. The sediment volume estimation margin of error was based on the vertical resolution of the seismic data used for surface interpolation (i.e. 0,1 m).

## 4 Results

### 4.1 Seabed morphology

The seabed features of the continental shelf of central-western Sardinia, including the distribution of rocky outcrops, sedimentary facies and seagrass meadows is showed in Fig. 6. The map was obtained integrating the geophysical and sedimentological data with the map of seagrass meadow of Sardinia island (Ministry of Environment Land and Sea, 2002), realised in 2002 by the Italian Ministry of Environment Land and Sea.

The outcropping and rocky basement were recognised mainly in the shelf sector facing the Sinis Peninsula and the Gulf of Oristano down to 150 m of depth. About 40 volcanic cones mounds, sub-circular in plain view (Fig.6), were recognised on the high-resolution DTM in a sector of approx. $40 \times 20$ km$^2$ (Conforti et al., 2016).

Sedimentary basins fill the depressions of the bedrock along the inner shelf. The backscatter data indicates a sharp acoustic contrast of the sedimentary deposits along the basins. The sector reported in Fig. 1 (150 km$^2$, 25 m to 140 m of depth) was considered the more interesting for the analysis of SSD. Backscatter data, calibrated with sedimentary data, revealed an alternation of low and high backscatter values that reveal the presence of finer sediments alternating with coarse sediments: this sedimentary pattern is associated with the presence of sorted bedforms (De Falco et al., 2015). Lower backscatter values allowed to identify the spatial limits of the finer sediments which corresponds to the transgressive sand layer deposit during the last sea level rise.

### 4.2 Submerged sand deposit

The features of sand deposits are shown in table1.The sedimentological analysis allowed to characterise the study area and sand reservoirs. Four textural and compositional groups were recognised: (i) muddy sands (CaCO3 66±8%), (ii) fine sands (CaCO3 63±17%); (iii) medium sands (CaCO3 43±29%) and (iv) sandy gravels(CaCO3 14±11%) (see De Falco et al., 2015 for the sedimentary facies classification). This subdivision is also of a spatial nature with the muddiest sediments located in the southernmost sector of the investigated area, the fine sands corresponding to the areas occupied by sorted bedforms and the medium sands located at the mouth of the Gulf of Oristano.

The muddy sands were not considered as potential SSD due to the high average mud content (24±11%). Fine and medium sands were fully characterised because those deposits were sampled by vibrocores. Consequently, the SSD reported in this work are referred to mixed fine and medium sand form whose geophysical and sedimentological data are available.

The stratigraphy of mixed fine and medium sand shows the presence of a few meters of acoustically transparent sedimentary cover, superimposed on a thicker layer formed by sandy gravel.

Twenty-five mixed fine and medium sand deposits have been detected for a total surface of 148 km$^2$(Fig. 5). The thickness reaches 4 meters but is mostly in the range of 1 to 2 meters. The total volume of the superficial sedimentary bodies has been estimated at 130,000 Mm$^3$.

An identification number (id_n) has been associated with each deposit (Table 1).

The deposits were classified in several categories to facilitate data management and description. Eight deposits category (named Type-n) are recognised based on the volumes value of the single deposit:

- The Type-1 (Volume <0.001Mm$^3$) consist of 4 deposits (id_01, id_02, id_03 and id_09) that have irrelevant volumes of less than 1000 m$^3$ each and thicknesses between 0.5m and 1m. These small deposits composed mainly by fine sand, have been mapped to complete the characterization SSD of the studied area.

- The Type-2 (Volume 0.01< 0.06Mm$^3$)is composed by 6 deposits (id_04, id_08, id_12, id_14, id_16 and id_21), has a total volume of 0.166 Mm$^3$ corresponding to 0.1% of the total volume detected and has thickness c and ranges from 0.50m to 2m. The deposits of this group cover an area of about 9.8 km$^2$ and consist mainly of fine and medium-fine sand.

- The Type-3 (Volume 0.10< 0.35 Mm$^3$) consists of 8 deposits (id_6, id_10, id_11, id_13, id_15, id_17, id_22 and id 23) that contain from 0.11 to 0.32 Mm$^3$ each, for a total of 1.41 Mm$^3$. They cover an area of about 3.6 km$^2$ and range from very fine to medium sand textural groups.

- The Type-4 (1.000 < 1.300 Mm$^3$) consists of three deposits (id_05, id_18 and id_20) made up of over 1 Mm$^3$ each, for a total of 3,424 Mm$^3$. They have a total surface of 6.6 km$^2$ and are characterised by thicknesses ranging from 0.50 m to 3 m. These deposits are characterised by fine sands.

- The Type-5 (3.00 < 4.00 Mm$^3$) consists of 2 deposits. The first (id_07) from 3.6 Mm$^3$ and the second from 3 Mm$^3$ (Id_19). The two deposits are developed for about 14.6 km$^2$ and are characterised by thicknesses ranging from 0.5m to 3m, consisting of fine and medium-fine sands.

- Type-6is represented by a single deposit (id_25) of 8.5 Mm$^3$, spread over an area of 16.8 km$^2$. The thickness of this deposit varies from 0.5 m to 2 m. The weaving class of sands from fine to very fine.

- Type-7 (>100 Mm$^3$) is represented by the biggest single deposit (id_24) which, with its 110 Mm$^3$, represents 85% of the total volumes measured. It covers an area of 104 km$^2$ and the thickness varies from 0.5 m to 4 m. The sands of this deposit belong to the fine and very fine dimensional classes.

**5 Data availability**

The dataset described in this study is publicly available, free of charge, from the PANGAEA data repository (Brambilla et al., 2018). The dataset presented is referred to the UTM 32N system WGS84 datum and is composed by all data needed to characterise the SSD, as well as the SSD themselves features.

**6 Discussion and conclusion**

Decision makers who face managing SSD need knowledge that can be derived from a complex dataset of geophysical and sedimentological data: in this work, we provided an example of organization of such knowledge, collected and delivered in a useful online fashion. A wide portion of the continental shelf of western Sardinian margin has been investigated. Along this area, several SSD were detected and characterised in terms of grainsize, stratigraphy, composition and available volumes.

This deposition area of SSD is located in a small structural basin limited by rocky outcrop, mainly formed by volcanic mounds of Early Pliocene (Conforti et al., 2016 and reference therein).

The mixed fine and medium sands derive from the mixing and winnowing of siliciclastic riverine sediments (De Falco et al., 2015b) with bioclastic sediments derived from marine benthic ecosystems (Posidonia oceanica and coralligenous assemblages) (De Falco et 2017). The fine and medium sand deposits are associated to submerged depositional terraces characterised by a distinct clinoform pattern and were considered as transgressive deposits related to the last sea level rise (Carboni et al., 1989, De Falco et al., 2015).

The compatibility of SSD with beach sediment of adjacent coastal area is limited to few sectors due to the great heterogeneity of beach sediment grain size and composition (De Falco et, al, 2017, 2014, 2003). Particularly beaches located along the coast ranging from the northern sector of the Gulf of Oristano to Cape Mannu, generally show a mixed composition with variable percentages of biogenic carbonate sediments. Some of these beaches are characterised by fine/medium sand which are fully compatible with the identified SDD.

This low degree of similarity can be attributed to the different processes which sorted beach and offshore sediments, the former characterised by an extremely variable composition and grain size (De Falco et al., 2003), due to specific local conditions (coastal morphology and geology, nearshore carbonate factories) which enhance strong coastal compartmentalization (Sanderson and Eliot 1999), the latter were probably selected under similar conditions driven by shelf hydrodynamics.

In addition, the evaluation of similarity must take into account other parameters (e.g. sediment colour, grain roundness, mineralogy) that we don't have evaluated in this work. Those sediment features are particularly important for pristine coastal sectors, such as the western Sardinia coast, where the specific beach sediment typology is a fundamental component of the coastal landscape.

The future policy of coastal area and the marine spatial plans will be driven by the expected sea level rise (Finkl and Khalil 2005; Armstrong et al., 2015;Barbanti et al., 2015; Brown et al., 2016 ) which will strongly influence the measures of protection against coastal flooding (Marchand et al., 2011; Mcglade et al., 2017).

Furthermore, as recommended by several projects funded by European Union Commission (Eurosion; Conscience, Micore) the recognition and classification of strategic sediment reservoirs that can be used as sediment supply for mitigation measures to contrast climate change are the basic concepts of coastal erosion management (Merchand et al., 2011).

In this context, the knowledge of SSD location and characterization will be fundamental to plan the sand reservoir exploitation, in order to plan coastal nourishment as protection against flooding.

This operations had to take in account the possible negative impact on the marine ecosystems and characterization and monitoring studies before, during and after the dredging, the sediment handling and the nourishment activities should be recommended to evaluate and minimize the potential impacts of these operations on the marine environment. For these reasons and to guarantee sustainable and long-term exploitation of not renewable resource as SSD, is important introducing management instruments focused on geological / physical characteristics of the sandy deposit, and the one concerning the environmental data acquired for the management and monitoring of the exploitation of the submerged marine sand resource (Nicoletti et al., 2018).

Well-curated, federated sources of dataset, organised in SDI, is strategic to provide a means to support of exploitation strategies and the projects that provide use and monitoring of dredging, both during and after mining.

sConcluding, this work summarizes a large dataset of geophysical and sedimentological data in order to map the spatial features of Submerged Sand Deposits as an information useful for coastal management at present and in future climate change scenarios.

**Author Contributions**

BW interpreted geophysical data, modelled the surfaces and thickness, mapped the seabed features, prepared the figures, wrote the manuscript and managed data and metadata handling on the online repository. CA acquired and processed the multibeam dataset and the seismic profiles and managed data and metadata handling on the online repository. SS, acquired and processed the multibeam, seismic and sedimentological data in various campaigns and contributed in writing the manuscript. CP and LS developed the interoperable system based on the software suite Geoinformation Enabling Toolkit StarterKit ® (GET-IT). DFG conceived and managed the study, processed data and contributed in writing the manuscript.

**Competing interests**

No potential conflict of interest was reported by the authors.

## Acknowledgements

The data of the present study were collected within the framework of the Magic project (Marine Geohazards along Italian Coasts) and the Ritmare project CNR (Sub-project SP4,Work-Package 1, Actions 1, 2). The activities about the interoperable system development activities have been funded by the Ritmare project. The authors are grateful to referees Jorge Guillen and Annamaria Correggiari for their helpful and all their good suggestions.

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

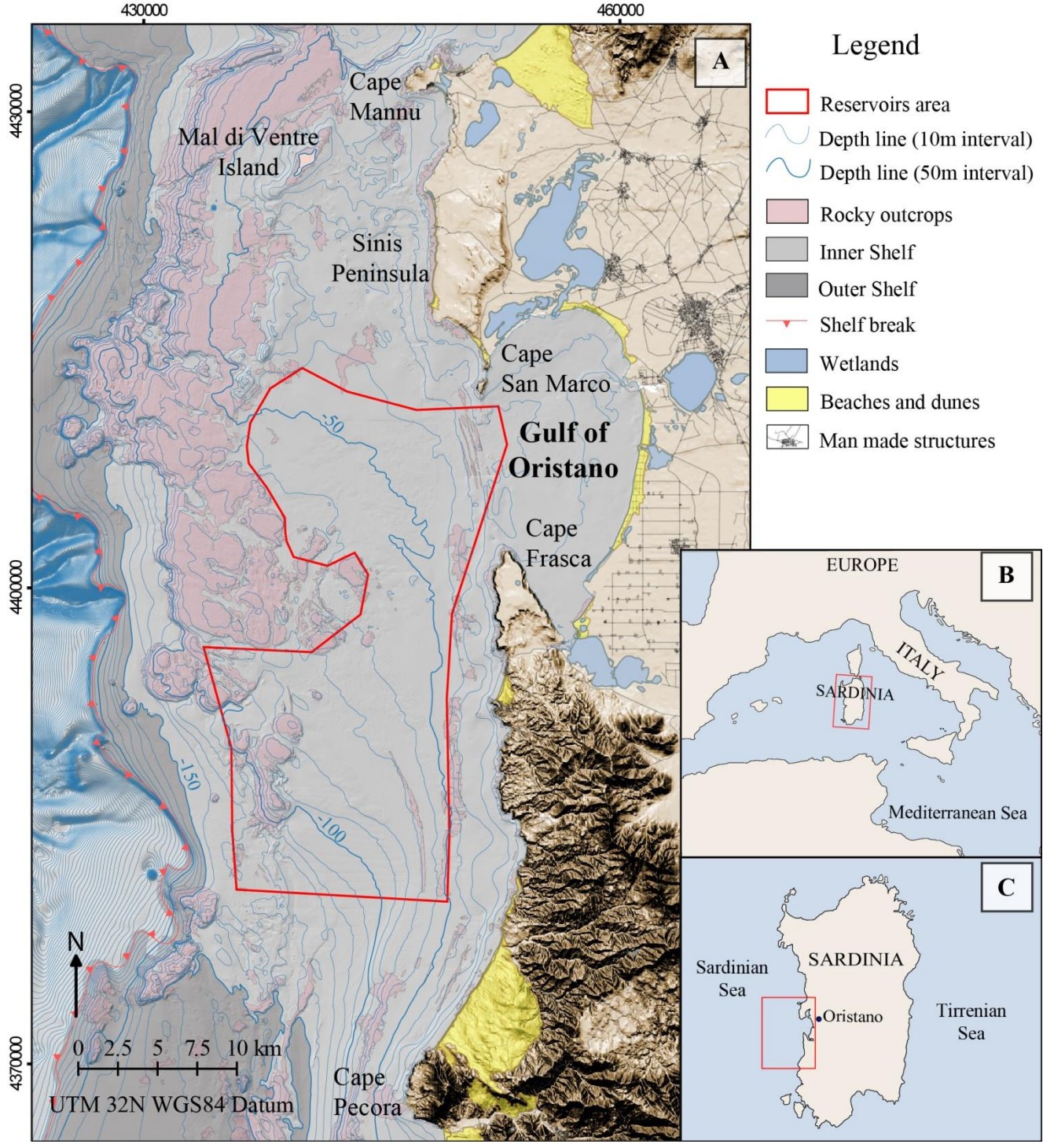

**Figure 1: (A) Location of the study area and schematic geomorphological map; (B) Geographic surroundings of the study area; (C) Regional map.**

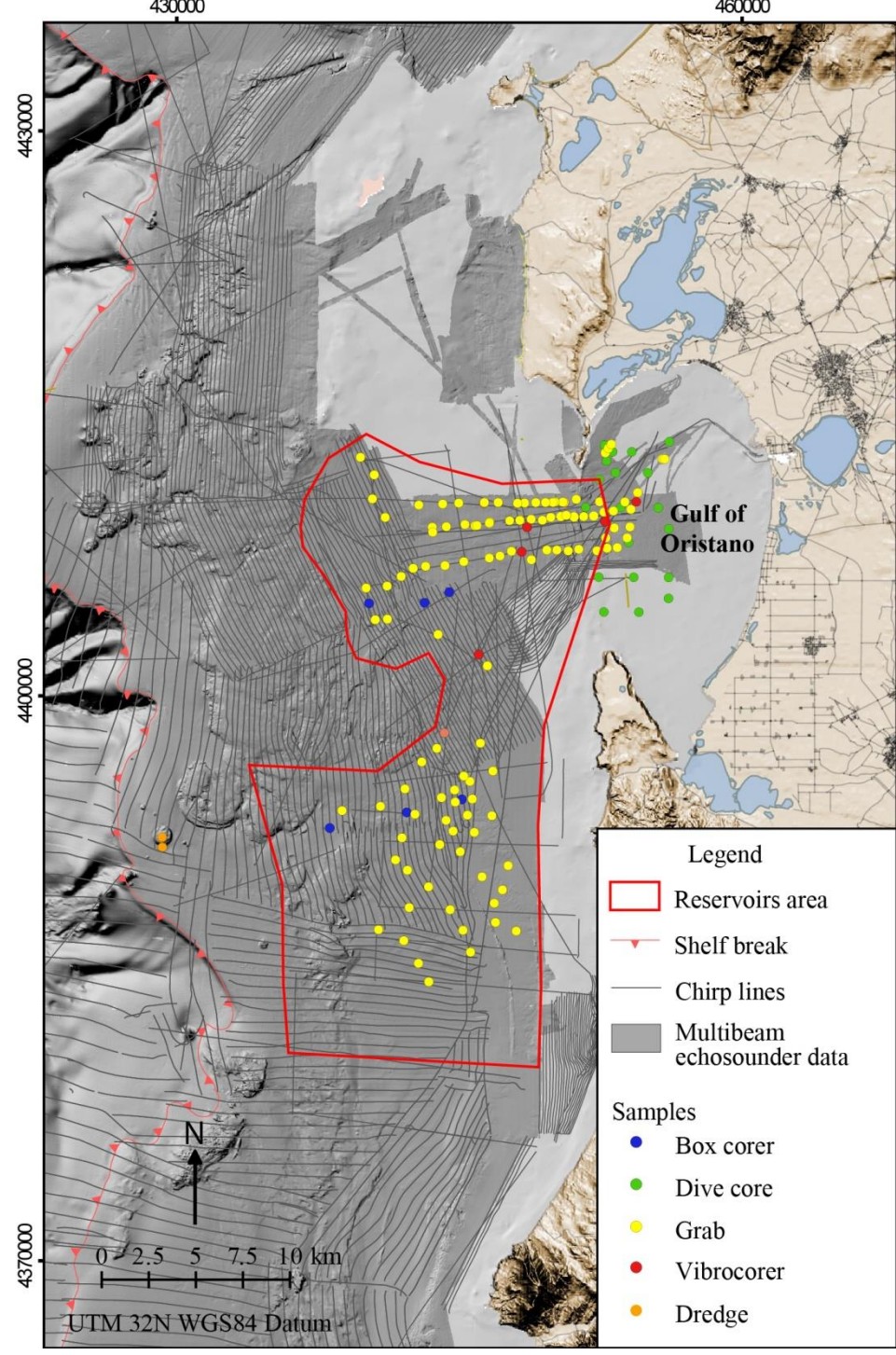

**Figure 2: Locations of acquired datasets.**

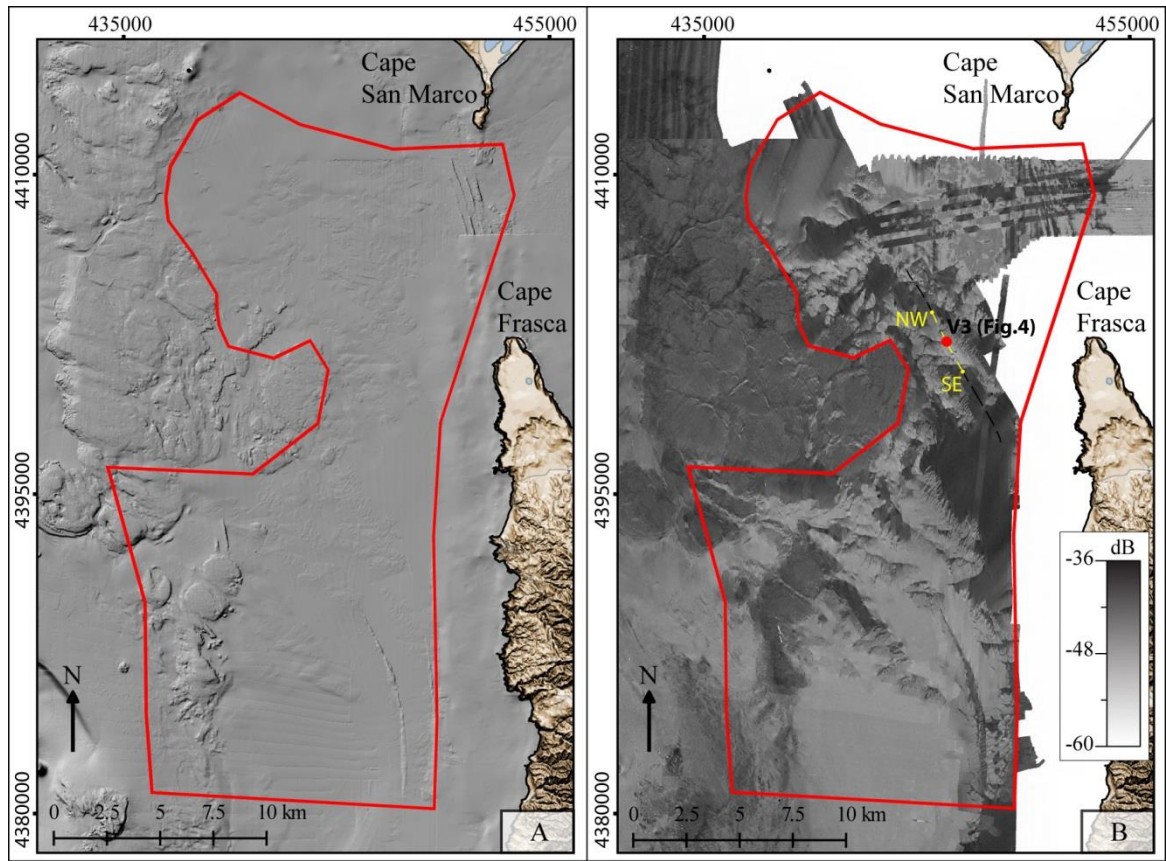

**Figure 3: A) Shaded relief map of digital terrain model of the study sector; (B) distribution of relative backscatter intensity values of the same sector.**

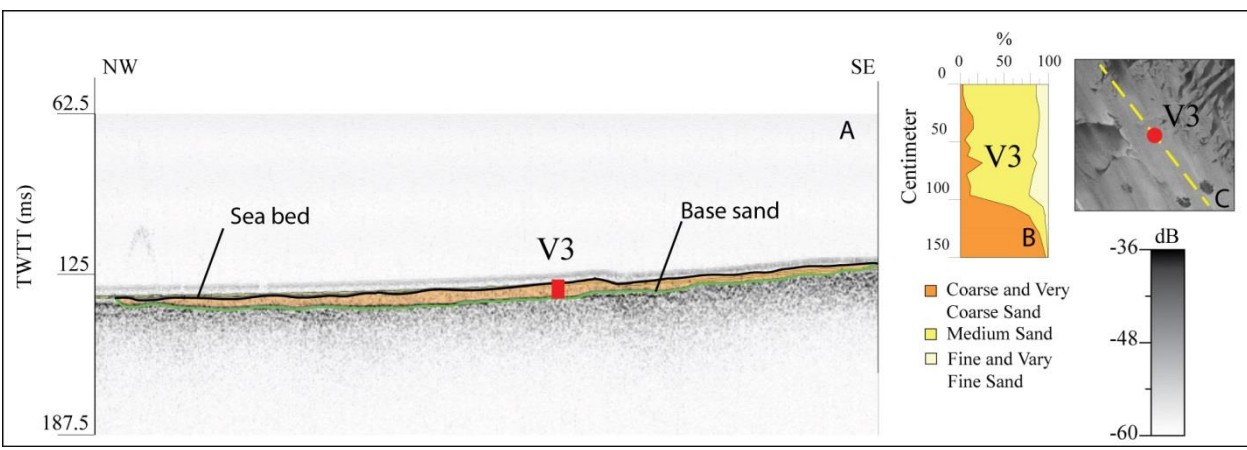

**Figure 4: (A) Example of digitization of the sand base in a chirp profile which is correlated with the core V3; (B) stratigraphic profile of the cores located on the chirp profile; (C) relative backscatter intensity value.**

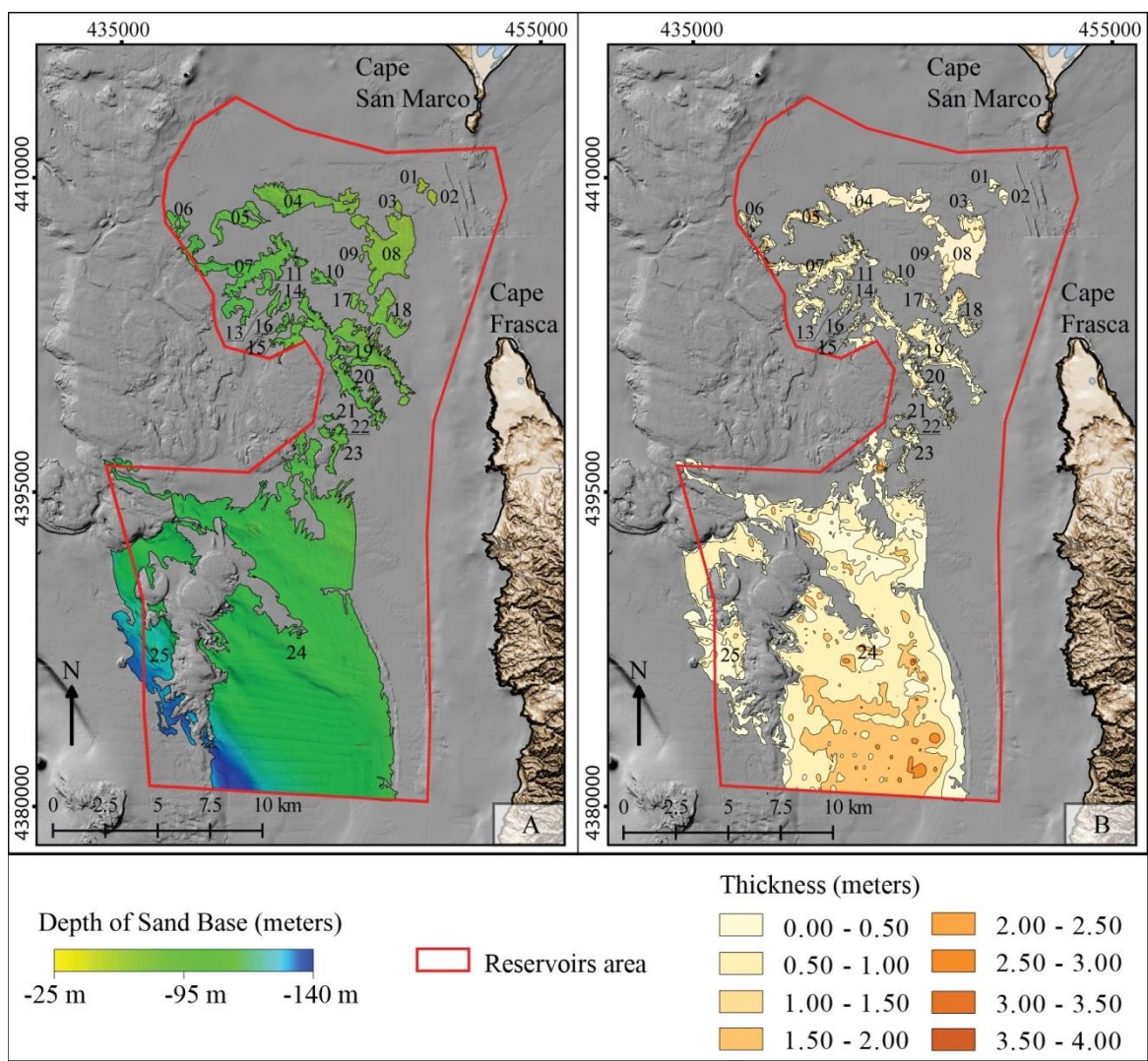

**Figure 5: (A) Digital terrain model of the sand base; (B) thickness map of Submerged Sand Deposits (SSD).**

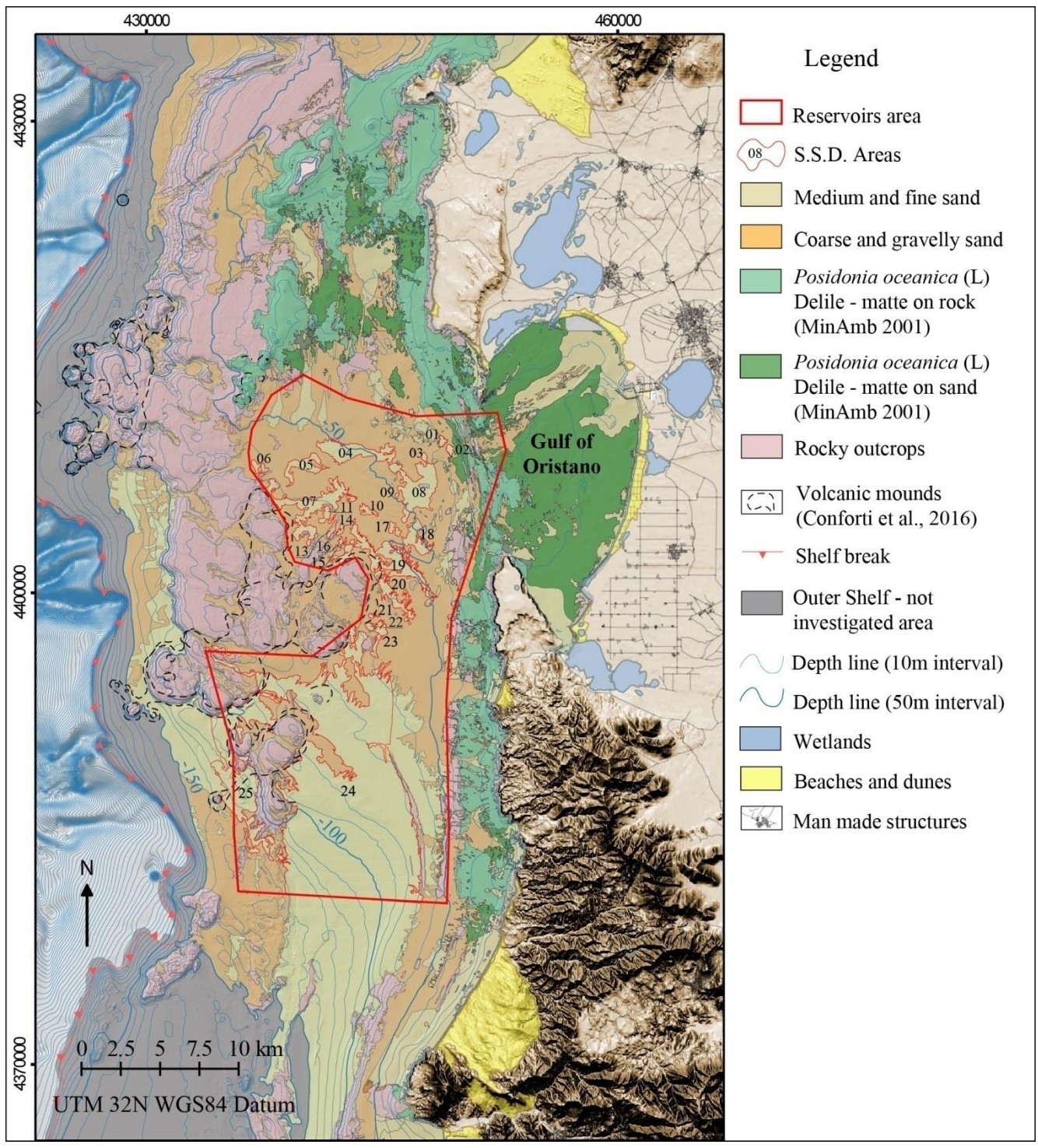

**Figure 6: Map of seabed features of the continental shelf of central-western Sardinia.**

| Name_id | Perimeter (km) | Surface (sq km) | Volume (m$^3$) | Sand class (Falk and Ward 1957) | Type |
|---|---|---|---|---|---|
| 01 | 2.7 | 0.24 | $<1 \times 10^3 \pm 100$ | Fine sand | 1 |
| 02 | 2.0 | 0.22 | $<1 \times 10^3 \pm 100$ | Fine sand | 1 |
| 03 | 2.0 | 0.16 | $<1 \times 10^3 \pm 100$ | Fine sand | 1 |
| 04 | 2.2 | 3.95 | $3.7 \times 10^3 \pm 370$ | Medium fine sand | 2 |
| 05 | 13.0 | 1.88 | $1285 \times 10^3 \pm 120000$ | Medium fine sand | 4 |
| 06 | 6.5 | 0.54 | $312 \times 10^3 \pm 30000$ | Medium fine sand | 3 |
| 07 | 45.8 | 5.95 | $3600 \times 10^3 \pm 360000$ | Medium fine sand | 5 |
| 08 | 20.0 | 5.54 | $16 \times 10^3 \pm 1600$ | Fine sand | 2 |
| 09 | 1.0 | 0.05 | $< 1 \times 10^3 \pm 100$ | Fine sand | 1 |
| 10 | 5.7 | 0.43 | $146 \times 10^3 \pm 14000$ | Fine sand | 3 |
| 11 | 3.5 | 0.41 | $161 \times 10^3 \pm 16000$ | Fine sand | 3 |
| 12 | 1.8 | 0.07 | $13 \times 10^3 \pm 1300$ | Fine sand | 2 |
| 13 | 3.5 | 0.32 | $115 \times 10^3 \pm 11500$ | Fine sand | 3 |
| 14 | 2.7 | 0.19 | $56 \times 10^3 \pm 5500$ | Fine sand | 2 |
| 15 | 3.6 | 0.27 | $110 \times 10^3 \pm 11000$ | Fine sand | 3 |
| 16 | 1.2 | 0.04 | $7.6 \times 10^3 \pm 760$ | Fine sand | 2 |
| 17 | 4.1 | 0.37 | $115 \times 10^3 \pm 11500$ | Fine sand | 3 |
| 18 | 12.1 | 1.83 | $1118 \times 10^3 \pm 111000$ | Fine sand | 4 |
| 19 | 65.5 | 6.53 | $3003 \times 10^3 \pm 300000$ | Fine sand | 5 |
| 20 | 29.1 | 2.12 | $1021 \times 10^3 \pm 10000$ | Fine sand | 4 |
| 21 | 2.6 | 0.13 | $36 \times 10^3 \pm 3600$ | Fine sand | 2 |
| 22 | 2.5 | 0.21 | $127 \times 10^3 \pm 12000$ | Fine sand | 3 |
| 23 | 11.0 | 0.96 | $322 \times 10^3 \pm 30000$ | Fine sand | 3 |
| 24 | 165.3 | 103.35 | $11038 \times 10^3 \pm 1104000$ | Fine / very fine sand | 7 |
| 25 | 72.5 | 16.32 | $8471 \times 10^3 \pm 800000$ | Fine / very fine sand | 6 |

**Table 1: Submerged Sand Deposit features.**

