# Peer review of "Dataset of Submerged Sand Deposits organised in an interoperable Spatial Data Infrastructure(Western Sardinia, Mediterranean Sea)"

_Earth System Science Data, 2018_

## Referee Comment (RC1) · Guillen (Referee) · 17 Jan 2019

Authors integrate an interesting dataset of morpho-sedimentological data of the W Sardinia continental shelf specifically oriented to the evaluation of sand reservoirs potentially available for coastal protection. I suggest two minor additions: a) at some point of the manuscript should be clarified that sand extraction on the continental shelf produces environmental impacts. Therefore, an accurate evaluation of these impacts should be required before extraction (and areas close to Posidonia meadows are specially sensitives); and b) to provide some information (or reference) about the sediment grain size of Sardinian beaches. Since the nourished sediment is usually coarser than

the original one in the beach, this information would provide a first approximation to how useful could be each sand deposit for coastal regeneration purposes. Other technical points: Page 3, lines 29-31, some reference? Page 4, lines 17-18, repeated sentence; Page 5, line 10, years in brackets; Page 8, lines 30-31, this information is repeated along the manuscript; Page 9, line 4, I suggest change "a tool useful in future climate change scenarios" to (or similar) "an information useful for coastal management at present and in future climate change scenarios"

---

## Author Comment (AC1) · 14 Feb 2019

Dear Reviewer, Thank you very much for your detailed comments and suggestions that helped to improve the manuscript and identify missing information that I now added in a revised manuscript.

---

## Referee Comment (RC2) · Annamaria Correggiari (Referee) · 20 Feb 2019

General comments The work highlights the results of the research for submerged sandy deposits on the Western Sardinia continental shelf potentially exploitable as a resource for coastal protection strategies and offers some tools to make their management more efficient. Regarding to this it would be important at least to cite, in the introduction, with some bibliographical references, the need to set up studies that evaluate the environmental impact of this type of activity on the seabed and the related marine ecosystem. Without this type of assessment it should not be possible to obtain permits to use the submerged sand resource. Specific Comments There

are several example of management projects and planning using Submerged Sand Deposits for renourishment of coastal areas but in Italy only Emilia-Romagna Region developed a collaboration with research to build a geodatabase (In_Sand, Correggiari et al. 2012 and 2016, last revised version) with a specific architecture, designed to the exploitation of the sand resource in terms of the sand characteristics, sand useful volumes available and the level of the lower limit of the sand deposit (referred by the authors as Sand Base). Another geodatabase has been implemented (Env_Sand, ref. Grande et al, 2015, result of a joint collaboration of CNR ISMAR and ISPRA in Ritmare Project CNR framework) with a complex architecture to organize all those data that must be collected in the environmental monitoring program before after and during the dreging activities of the Submerged Sand Deposits. (Nicoletti et al 2018; MATTM-Regioni, 2018). Reference* Nicoletti L., La Valle P., Paganelli D., Lattanzi L., La Porta B., Targusi M., Lisi I., Loia M., Maggi C., Pazzini A., Proietti R., Gabellini M. (2018) - Aspetti ambientali del dragaggio di sabbie relitte a fini di ripascimento: protocollo di monitoraggio per l'area di dragaggio. ISPRA, Manuali e Linee Guida 172/2018, 35 pp. Technical corrections Add those technical corrections to the ones highlited by RC1: Page 2 line 10 EC Projects: Coscience and Micore should be described like Eurosion Page 2 line 31: better use "Ministry of Environment Land and Sea" as is written in the english version of its italian name. Page 2 line 32: (the updated reference, with "Eurosion, 2004" is: "MATTM – Regioni, 2018") * the complete ref is a book in italian available on the site: http://www.erosionecostiera.isprambiente.it/files/linee-guida-nazionali/TNEC_LineeGuidaerosionecostiera_2018.pdf *Ref: MATTM-Regioni, 2018. Linee Guida per la Difesa della Costa dai fenomeni di Erosione e dagli effetti dei Cambiamenti climatici. Versione 2018-Documento elaborato dal Tavolo Nazionale sull'Erosione Costiera MATTM-Regioni con il coordinamentotecnico di ISPRA, 305pp Page 3 line 1-3 and line 9-10: Page 3 line 1-3 and line 9-10: I would like to encourage the authors to better highlight the importance of introducing this type of instruments, the ones focused on geological / physical characteristics of the sandy deposit, and the one concerning the environmental data acquired for the management and monitoring

of the exploitation of the submerged marine sand resource. Use the right reference papers for both the types. (in_Sand: Correggiari et al., 2016 and Env_Sand: Grande et al 2015) Page 3 line 9 instead of "marine space" better "maritime space" Page 6 line 2 add Correggiari et al. 2016 Page 6 line 12: better use "Ministry of Environment Land and Sea" as is written in the english version of its italian name. Page 13 line 12 instead of 2015 the ref is 2005 Page 19 Table 1 Could be useful to add square km to the Submerged Sand Deposit features

---

## Author Response (AR1)

**Response to reviewer#1**

Dear Jorge Guillen, thank you very much for your detailed comments and suggestions.

Here below our replies, point by point, to the comments of the reviewer#1. The comments from the reviewer are in bold, while our response is in normal font.

**Comment a) at some point of the manuscript should be clarified that sand extraction on the continental shelf produces environmental impacts. Therefore, an accurate evaluation of these impacts should be required before extraction (and areas close to Posidonia meadows are specially sensitives).**

**We added in the "Introduction":**

[revised manuscript text omitted]

**Comment: page 4, lines 17-18, repeated sentence.** We have deleted the sentence (page 5, line 21 of the revised manuscript)**.**

**Comment: page 5, line 10, years in brackets.** We added the brackets (page 6, line 14 of the revised manuscript).

**Comment: page 8, lines 30-31, this information is repeated along the manuscript.** We have deleted the redundant information (page 10, line 26 of the revised manuscript).

**Comment: page 9, line 4, I suggest change "a tool useful in future climate change scenarios" to (or similar) "an information useful for coastal management at present and in future climate change scenarios".** We changed the phrase (page 10, line 31 of the revised manuscript).

**Response to reviewer#2**

Dear Annamaria Correggiari, thanks for the time you dedicated to reviewing this paper and all your good suggestions. The comments from the reviewer are in bold, while our response is in normal font.

**Comment: "…it would be important at least to cite, in the introduction, with some bibliographical references, the need to set up studies that evaluate the environmental impact of this type of activity on the seabed and the related marine ecosystem. Without this type of assessment, it should not be possible to obtain permits to use the submerged sand resource. Specific Comments There are several examples of management projects and planning using Submerged Sand Deposits for nourishment of coastal areas but in Italy only Emilia-Romagna Region developed a collaboration with research to build a geodatabase (In_Sand, Correggiari et al. 2012 and 2016, last revised version) with a specific architecture, designed to the exploitation of the sand resource in terms of the sand characteristics, sand useful volumes available and the level of the lower limit of the sand deposit (referred by the authors as Sand Base). Another geodatabase has been implemented (Env_Sand, ref. Grande et al, 2015, result of a joint collaboration of CNR ISMAR and ISPRA in Ritmare Project CNR framework) with a complex architecture to organize all those data that must be collected in the environmental monitoring program before after and during the dredging activities of the Submerged Sand Deposits."**

**We added in the "Introduction" (page 3, lines 5-14 of the revised manuscript):** There are several examples all around the world on the utilization of SSD for the nourishment projects (Finkl et al 2005). In Italy, the Emilia-Romagna Region developed a geodatabase named IN_SAND, that help the managers to plan the SSD exploitation (Correggiari et al. 2012 and 2016). It is also known that the dredging operation can negatively affect the marine ecosystems and in particular the sessile benthic communities, (Rogers, 1990; Desprez, 2000; Erftemeijer et al., 2012; Fraser et al., 2017). For this reason, accurate procedures aimed to minimize the impact of nourishment operations (including dredging) were adopted by several countries and must to be required when a nourishment project is developed (Finkl et al., 2005; Radermacher et al., 2017). Along the Mediterranean Sea, this operation can affect the *Posidonia oceanica* meadows. In Italy, for this reason, ISPRA and CNR develops the ENV_SAND database (Grande et al., 2015) aimed to collect the data acquired during the monitoring programme of a nourishment project, including the dredging of the SSD (Nicoletti et al., 2018; MATTM-Regioni, 2018).

**Also, we added in the "Discussion and conclusion" (page 10, lines 8-19 of the revised manuscript):** Furthermore, as recommended by several projects funded by European Union Commission (Eurosion; Conscience, Micore) the recognition and classification of strategic sediment reservoirs that can be used as sediment supply for mitigation measures to contrast climate change are the basic concepts of coastal erosion management (Merchand et al., 2011). In this context, the knowledge of SSD location and characterization will be fundamental to plan the sand reservoir exploitation, in order to plan coastal nourishment as protection against flooding. These operations had to take in account the possible negative impact on the marine ecosystems and characterization and monitoring studies before, during and after the dredging, the sediment handling and the nourishment activities should be recommended to evaluate and minimize the potential impacts of these operations on the marine environment. For these reasons and to guarantee sustainable and long-term exploitation of not renewable resource as SSD, is important introducing management instruments focused on geological / physical characteristics of the sandy deposit, and the one concerning the environmental data acquired for the management and monitoring of the exploitation of the submerged marine sand resource (Nicoletti et al., 2018).

**Other technical points:**

**Comment: page 2 line 10 EC Projects -Conscience and Micore should be described like Eurosion.**

**We added in the "Introduction", (page 2, lines 15-18 of the revised manuscript):** The Eurosion concepts were applied in the CONSCIENCE Project that includes on the sediment cell also the sediment reservoirs that can act as a source of sediment (Van Rijn, 2010). These aspects are also discussed on MICORE project that stressed on the application of nourishment projects to contrast the climate change effects, in particular, the extreme storms and the sea level rise (Ciavola et al., 2011).

**Comment: page 2 line 31: better use "Ministry of Environment Land and Sea" as is written in the English version of its Italian name.** We used "Ministry of Environment Land and Sea" instead of "Ministero dell'Ambiente e della Tutela del Territorio" in several pages of the revised manuscript:

- page 3, line 2 and line 28;
- page 7, line 14 and 15

**Comment: page 2 line 32 – the updated reference, with "Eurosion, 2004" is: "MATTM – Regioni, 2018".** We added the reference on page 3, line 3 of the revised manuscript.

**Comment: page 3 line 1-3 and line 9-10: I would like to encourage the authors to better highlight the importance of introducing this type of instruments, the ones focused on geological / physical characteristics of the sandy deposit, and the one concerning the environmental data acquired for the management and monitoring of the exploitation of the submerged marine sand resource.**

**What added in "Introduction" (page 3, lines 14-18 of the revised manuscript):** The introduction of these geodatabases provide instruments that could be very helpful on the management of the SSD not only in terms of sand volumes but useful to (i) support the procedures of environmental impact assessment, (ii) monitor the marine environmental condition during the activities and (iii) manage the impact of dredging and nourishment activities on marine ecosystems (Erftemeijeret al., 2012; Fraser et al., 2017; Nicoletti et al., 2018; MATTM-Regioni, 2018).

**Comment: use the right reference papers for both the types. (in_Sand: Correggiari et al., 2016 and Env_Sand: Grande et al 2015).** We used the right reference papers on page 3, line 28 of the revised manuscript.

**Comment: page 3 line 9 instead of "marine space" better "maritime space".** We have changed the word indicated on page 3, line 27 of the revised manuscript.

**Comment: page 6 line 2 add Correggiari et al. 2016.** We added the reference on page 7, line 5 of the revised manuscript.

**Comment: page 13 line 12instead of 2015 the ref is 2005.** We have changed the wrong reference how to indicated by the reviewer in page 15, line 29 of the revised manuscript.

**Comment: page 19, Table 1 - Could be useful to add square km to the Submerged Sand Deposit features.** We added the "Surface" column (sq km) in the table of Submerged Sand Deposit features.

**Dataset of Submerged Sand Deposits organised in an interoperable Spatial Data Infrastructure(Western Sardinia, Mediterranean Sea)**

Walter Brambilla[1], Alessandro Conforti[1], Simone Simeone[1], Paola Carrara[2], Simone Lanucara[2] and Giovanni De Falco[1]

[1]Istituto per lo studio degli impatti Antropici e Sostenibilità in ambiente marino del CNR, Oristano, Italy

[2] Istituto per il Rilevamento Elettromagnetico dell'Ambiente CNR, Milano, Italy

*Correspondence to*: Walter Brambilla (walter.brambilla@iamc.cnr.it)

**Abstract.** The expected global sea level rise by the year 2100 will determine an adaptation of the whole coastal system and the land retreat of the shoreline. Future scenarios coupled with the improvement of mining technologies will favour increased exploitation of sand deposits for nourishments, especially for urban beaches and sandy coasts with lowlands behind. Objective of the work is to provide useful tools to support planning actions in the management of sand deposits located on the continental shelf of western Sardinia (western Mediterranean Sea). The work has been realised through the integration of data and information collected during several projects. Available data consist of morpho-bathymetric data (multibeam) associated with morphoacoustic (backscatter) data, collected in the depth range -25 to-700 m. Extensive coverage of high-resolution seismic profiles (Chirp 3.5 kHz) have been acquired along the continental shelf. Also, surface sediment samples(Van Veen grab and box corer) and vibrocores have been collected. These data allow mapping of the submerged sand deposits with the determination of their thickness and volumes, and their sedimentological characteristics. Furthermore, it is possible to map the seabed geomorphological features of the continental shelf of western Sardinia. All the available data (doi: https://doi.pangaea.de/10.1594/PANGAEA.895430) have been integrated and organised in a geo-database implemented through a GIS and the software suite Geoinformation Enabling Toolkit StarterKit ® (GET-IT), developed by researchers of the Italian National Research Council for RITMARE project. GET-IT facilitates the creation of distributed nodes of an interoperable Spatial Data Infrastructure (SDI) and enables unskilled researchers from various scientific domains to create their own Open Geospatial Consortium (OGC) standard services for distributing geospatial data, observations and metadata of sensors and datasets.

Data distribution through standard services follows the guidelines of the European Directive INSPIRE (DIRECTIVE 2007/2/EC); in particular, standard metadata describe each map level, containing identifiers such as data type, origin, property, quality, processing processes to foster data searching and quality assessment.

**Copyright statement**

The work presented here is provided under the terms of the Creative Commons License 4.0 (CC BY 4.0).

[revised manuscript text omitted]